# Relationships between Health Education, Health Behaviors, and Health Status among Migrants in China: A Cross-Sectional Study Based on the China Migrant Dynamic Survey

**DOI:** 10.3390/healthcare11121768

**Published:** 2023-06-15

**Authors:** Minji Kim, Hai Gu

**Affiliations:** 1School of Government, Nanjing University, Nanjing 210023, China; 2Center for Health Policy and Management Research, Nanjing University, Nanjing 210023, China

**Keywords:** migrant population, health education, health behavior, health status, China Migrant Dynamic Survey, logistic regression analysis, structural equation model

## Abstract

Managing the health of migrants has become a crucial aspect of promoting social harmony and cohesion in China. This study investigates the impact of public health education on the health status of migrants in China using cross-sectional data from the China Migrants Dynamic Survey 2017. A total of 169,989 migrants in China were selected as samples for empirical test. Data were analyzed using descriptive statistics, logistic regression, and the structural equation model. The findings show that health education significantly influences the health status of migrants in China. Specifically, health education related to occupational diseases, venereal diseases/AIDS, and self-rescue in public emergencies had a significant positive impact on migrants’ health, while health education regarding chronic diseases had a significant negative impact. Health education delivered through lectures and bulletin boards had a significant positive impact on migrants’ health, but online education had a significant negative effect on the health status of migrants. The effects of health education differ by gender and age, with a stronger positive impact on female migrants and elderly migrants aged 60 and above. The mediating effect of health behaviors was significant only in the total effect. In conclusion, health education can effectively enhance the health status of migrants in China by modifying their health behaviors.

## 1. Introduction

After implementing reform and opening up policies, China experienced rapid urbanization, which attracted surplus rural labor and facilitated the growth of internal migration. *Liudong renkou* is a term used in China to refer to internal migrants who move between cities and provinces in search of better opportunities, following a temporary and circular pattern [1]. These migrants have emerged as a consequence of the country’s fast-paced industrialization and urbanization, and have played a significant role in China’s economic and social advancements [2]. According to the 7th National Population Census conducted in 2020 [3], the number of migrants in China has increased to 376 million, accounting for approximately 27% of the total population, a significant increase from the 200 million recorded in the 2010s. Due to the substantial size of the migrant population in China, providing healthcare to this group has become an indispensable task that cannot be overlooked.

Migrants are at a higher risk of developing health issues due to poor working conditions and a lack of social security and support. These issues can lead to various health problems, such as infectious diseases, occupational diseases, reproductive health issues, and mental health disorders [4,5,6]. In addition, some infectious diseases can also affect the families of migrants and the health of residents in both the inflow and outflow areas [7]. Furthermore, health inequality is a significant problem for migrants, who are often excluded from the medical service benefits provided to residents due to the household registration system [8,9,10]. Although the Chinese government has implemented policies to improve the availability of public healthcare services for migrants and reduce health risks, the utilization rate of these services remains generally low. This can be attributed to the relatively weak health consciousness of migrants, as highlighted in the National Health Commission’s 2018 report on the development of migrants in China [11]. Through the experience of large-scale infectious diseases such as coronavirus disease 2019, it has been observed that individual health literacy and hygiene behavior play crucial roles in controlling the transmission of the virus, especially in areas with high population mobility during outbreaks [12,13]. Therefore, improving the health literacy and health behavior of the migrant population is a significant challenge. To address this issue, it is essential to promote scientific health knowledge, raise awareness about health, and advocate for healthy lifestyles among migrants in China. These measures are essential to address the urgent health requirements of this population and reduce the health risks associated with insufficient health services, while ensuring social equity.

Health education is a crucial means of improving the health of migrants. According to Nutbeam and Kickbusch’s [14] health promotion glossary, health education refers to learning that includes some sort of communication and is intended to increase health literacy, knowledge, and life skills that are favorable to both individual and community health. Health behaviors refer to “personal attributes such as beliefs, expectations, motives, values, perceptions, and other cognitive elements; personality characteristics, including affective and emotional states and traits; and overt behavior patterns, actions, and habits that relate to health maintenance, to health restoration and to health improvement” [15] (p. 169). Health status is commonly defined as individuals’ self-perceived health [16]. This multidimensional concept encompasses various aspects, such as physical, cognitive, emotional, and social well-being, as well as the presence or absence of disabilities [17]. Health surveys often capture these dimensions to assess an individual’s overall health status. It is often referred to as self-rated health status and has consistently shown its significance in predicting important health statuses, including mortality and morbidity [18]. By increasing health-related knowledge and promoting health literacy, health education can influence health behaviors and help individuals to address their social determinants of health, ultimately leading to positive health status [19,20]. Therefore, it is essential to study the impact of health education on the health of migrants. However, previous studies have primarily focused on the relationship between health knowledge and health behaviors, without expanding the discussion to the mechanism by which health knowledge affects the health status of migrants. As a result, there is a gap in knowledge regarding the mechanism by which health knowledge influences the health status of migrants.

This study aims to expand on previous studies by examining the causal relationship between public health education and the health status of migrants in China. Specifically, this study aims to investigate: (1) how health education affects the health status of migrants and (2) whether health behaviors mediate the relationship between health education and health status of migrants. Through a comprehensive analysis of these research questions, this study intends to provide valuable suggestions for enhancing the effectiveness of public health education methods.

## 2. Literature Review

In order to establish the rationale for investigating the effects of public health education on the health status of migrants in China and the underlying mechanisms, this study conducted a comprehensive review of relevant previous studies. Methodologically, a systematic review approach was not adopted, but rather, a thorough review of select representative literature was undertaken. Identified limitations from a limited number of previous studies were addressed, and a theoretical foundation was established to analyze the relationship mechanism between health education and the health status of migrants in China.

### 2.1. Limitations of the Previous Studies

Health education is a crucial tool to improve the health of migrants by promoting health literacy and positively influencing health behaviors. This relationship between health education and health status is particularly relevant for migrants in China, who often face limited access to healthcare due to the household registration system. Migration can increase the vulnerability of migrants to both communicable and non-communicable diseases. Castelli and Sulis [21] noted that infectious diseases continue to be a significant cause of mortality among migrants. Regarding non-communicable diseases, Davies, Basten, and Frattini [22] indicated that migrants often engage in unskilled labor, which may lead to poor working conditions and an increased risk of occupational diseases and injuries. Additionally, factors such as poverty, experiences of sexual abuse (particularly among female domestic workers), a lack of social support, stigma, and discrimination may contribute to the incidence of mental health issues such as depression. Finally, Ebrahim and Smeeth [23] observed that changes in lifestyle and dietary habits following migration may contribute to a higher incidence of obesity and diabetes.

Despite being vulnerable to communicable diseases and non-communicable diseases, migrants have been found to possess limited knowledge and risk awareness regarding these diseases [24]. Despite the need for research on health knowledge of migrants, previous studies have primarily focused on the pattern of health service utilization (medical services) and its relationship with migrants’ health status [25,26,27,28]. Further research is needed to explore the potential impact of health education on Chinese migrants’ health status, especially in addressing the health inequalities they often face.

While there have been limited studies on health education for migrants in China, previous studies have primarily focused on the impact of health knowledge on health behavior [29,30,31]. Previous studies have not expanded the discussion to how health education and/or health knowledge affect health status. Yu et al. [32] studied the causal relationship between health literacy and the health status of migrants in China, and they set health behavior (i.e., health service utilization) as a mediating factor. They found a positive correlation between health literacy, health service utilization, and health status. Their analysis of mediating effects suggests that health behaviors have a partial mediating effect between health literacy and health status. However, more empirical studies are needed to back this effect and unveil the mechanism by which health knowledge affects the health status of migrants in China.

### 2.2. The Impacts of Health Knowledge and Health Behaviors on the Health of Migrants

The enhancement of health literacy through health education pertains to an individual’s or group’s heightened capacity to comprehend health-related information and to make informed decisions regarding the use of appropriate healthcare services and the adoption of healthy lifestyle practices. Such advancements in health literacy have the potential to facilitate positive health status, including medical advancements, the adoption of healthy lifestyles, and even the initiation of social movements aimed at promoting institutional change. These outcomes are rooted in health behavior change, which occurs through a series of interventions and education efforts aimed at improving the health literacy of individuals and groups. 

The influence of health knowledge on health status is widely recognized in the field. However, it is important to note that this relationship is not direct, as health literacy does not serve as a direct determinant of health status. Rather, the improvement of health literacy through health education and knowledge dissemination is generally considered an effective approach to promote health improvement via a mediating process known as health behavior change. Scholars have conducted comprehensive reviews of relevant theories and developed health literacy frameworks to comprehend the mechanisms underlying the relationship between health literacy and health status. As a notable example, Paasche-Orlow and Wolf [33] considered established health behavior theories, such as the Health Belief Model (HBM hereafter) and the Theory of Planned Behavior (TPB hereafter), as well as the causal pathways between health literacy and health status, and demonstrated that health behaviors can serve as a mediator between health literacy and health status.

In the field of health education, effective communication is essential, and scholars have explored the use of effective communication tools. For example, Schiavo [34] systematized theories from different fields, such as media, marketing, and sociology, to identify effective health communication practices. However, as pointed out by Thomas, Chase, and Aggleton [35], it is not well understood what types of health education tools work best for migrants with different life experiences. Zhang et al. [36] observed the effects of a health education program targeting Chinese migrant women working in overseas-branded factories, which led to improved health literacy and behavior changes. The program used brochures and booklets from China’s Center for Disease Control and free physicals provided by local clinics as educational media. Mendelsohn et al. [37] described the design process of an HIV and sexually transmitted infection (STI) education program for migrant construction workers in Shanghai, which was implemented through the use of pamphlets, videos, and individual counseling, and was effective in promoting HIV and STI knowledge in migrant construction workers. Guo et al. [38] showed that online health education using platforms such as WeChat, mobile applications, and websites had a positive effect on improving the health literacy of migrants in China. However, further investigation is required to explore the causal relationships between effective communication media and the health status of migrants in the context of health education.

This study aims to investigate the causal relationship between health education, health behaviors, and the health status of migrants in China, utilizing the ‘China Migrants Dynamic Survey (CMDS hereafter)’ database. Figure 1 illustrates the analytical framework used in this study. This paper is structured as follows: Section 3 outlines the research methodology, Section 4 presents the findings, Section 5 includes a discussion of the results, and Section 6 offers the conclusions.

## 3. Materials and Methods

### 3.1. Data Source

The CMDS conducted by the National Health Commission of China from 2009 to 2018 aimed to gain insights into various aspects of the migrant population, such as survival and development status, migration trends, the utilization of public health services, and the management of family planning services. This study focused on analyzing the impact of health education on the health status of migrants in China, while also exploring the underlying mechanisms involved. The survey, focusing specifically on public health education for the migrants, was conducted only in 2017. The 2017 CMDS dataset provided a wealth of information on public health education services, health behaviors, and health status among migrants in China. The 2017 CMDS coding system allows for the accurate identification of causal relationships between health education and the health status of migrants, making the dataset suitable for this study’s research objectives. Consequently, the 2017 CMDS dataset was selected for our analysis.

The survey excluded individuals who temporarily stayed at places such as stations, docks, airports, hotels, and hospitals during the survey. The survey used probability proportional to size (PPS) sampling, which was a method of sampling from a finite population in which the probability of unit selection was proportional to size. The survey employed a three-stage sampling method. In the first stage, township-level units were selected from the 32 provincial-level administrative units. In the second stage, village committees were selected from the chosen township-level units. In the final stage, the migrant population suitable for the survey was selected from the identified village committee.

### 3.2. Study Population

The 2017 CMDS survey covered 32 provincial-level administrative units across the country, with a target sample size of 169,989. The 2017 CMDS data consisted of individual and community levels. Individual-level data mainly include family members, economic conditions, health status, the utilization of public services, and the work and education status of respondents. The individual-level respondents were those who had resided in the sample location for more than one month and were not residents of the sample district (county, city), but were over 15 years old and migrated for work and life purposes. Their migration range encompasses cross-provincial, cross-city, cross-county, and cross-border movements. Figure 2 represents a flowchart illustrating the characteristics of the study population’s migration range. As shown in Figure 2, cross-provincial migration accounts for the highest number of individuals, with 83,790 people, followed by cross-city migration with 56,017 individuals, while cross-border migration has no recorded individuals. Migrants involved in cross-county and cross-border movements were excluded from this study. For convenience purposes, migrants aged 15 years and above but under 20 years, who constituted a relatively small sample size, were also excluded. The collected sample data were analyzed using IBM SPSS Statistics version 29.

### 3.3. Measures

#### 3.3.1. Dependent Variables

The self-rated health status (SRHS hereafter) is frequently examined in epidemiological surveys to assess individuals’ overall well-being in terms of social, biological, and psychological health [39,40]. SRHS serves as an indicator that can be applied across various contexts and can be used as a proxy for actual health status [41]. Therefore, SRHS can be utilized as a measure to assess the health status of migrants in China. Respondents were asked to categorize their health status as ‘unable to take care of myself’, ‘unhealthy’, ‘basically healthy’, or ‘healthy’ in the CMDS questionnaire. The dependent variables were assigned a score of 0–1 (0 = unhealthy and unable to take care of myself, 1 = healthy and basically healthy).

#### 3.3.2. Independent Variables

In our study, we included a question from the CMDS questionnaire that asked respondents if they had participated in any of the nine types of public health education programs during the past year. These programs represent National Basic Public Health Services offered by the Chinese government to the entire population through community-based initiatives implemented in local villages since 2009 [42]. As a result, the program themes and content remain consistent across regions. We considered this health education variable as continuous, with higher values indicating a greater number of attended health education programs. This approach enabled us to examine the correlation between health education and health status among the migrant population. For the second major independent variable, we selected seven out of the nine types of health education programs based on the existing research literature and knowledge of the most prevalent health issues among the vulnerable migrant population in China. These seven types of health education included occupational disease, venereal disease/AIDS, reproductive health, mental health, chronic health, maternal and child health, and self-rescue in public emergencies [43,44,45,46,47,48,49]. Each type was transformed into a dummy variable to investigate the relationship between the type of disease prevention education and the health status of migrants. The third major independent variable was the type of education medium. There were five types of education medium, including lectures, publicity material, bulletin boards, public consultations, and online education. Each type was transformed into a dummy variable to explore the association between the type of education medium and the health status of migrants.

#### 3.3.3. Mediation Variables and Control Variables

The mediation variables in this study are health behaviors that consist of two main aspects: medical-seeking behavior and hygiene behavior. In the health and public services and social integration module of the CMDS database, the questionnaire asked migrants about their attitudes towards medical-seeking behaviors and hygiene behaviors from their perspective. To determine the medical-seeking behavior of migrants, the questionnaire asked, ‘Do you seek treatment when symptoms of infectious diseases appear?’ The response of migrants was then categorized as 0 for ‘not treated’ and 1 for ‘treated.’ Similarly, to determine the hygiene behavior of migrants, the questionnaire asked, ‘Do you agree that your hygiene habits are quite different from those of the local people?’ The degree of agreement was categorized as 0 for ‘agree’ and 1 for ‘disagree’.

To control for other factors that may influence the health status of migrants, this study classified control variables into three categories: demographic characteristics, socio-economic status, and health service publicity, drawing on previous studies. Demographic characteristics and socio-economic status are recognized as important determinants of health status, as they can have a significant impact on an individual’s access to healthcare, exposure to environmental hazards, and ability to make healthy lifestyle choices, among other factors [50]. Demographic characteristics include gender (0 = female, 1 = male), age, educational level, marital status (0 = no, 1 = yes), household registration (0 = others, 1 = agricultural household registration), migration range (1 = across cities within a province, 2 = across provinces), and current residence. More precisely, the age groups were divided into 20–29, 30–39, 40–49, 50–59, 60–69, and 70 years of age and older. Education level reflected the highest education level of the migrants. Responses were given on a seven-point rating scale, ranging from 1 to 4 (1 = have not attended school, 2 = middle school, 3 = high school, and 4 = college and above). Current residence reflected the geographical distribution of the migrants’ residence province/city, divided into three categories: 1 (east), 2 (middle), and 3 (west). Socio-economic status encompassed factors, such as availability of medical insurance (0 = no, 1 = yes), monthly household income, and employment status (0 = unemployed, 1 = employed). More precisely, monthly household income was divided into four categories: 1 (CNY −90,000 to 4000), 2 (CNY 4001 to 6000), 3 (CNY 6001 to 8000), and 4 (CNY 8001 to 200,000). The health service publicity variable was constructed based on responses to the question, ‘Have you heard of the National Basic Public Health Services?’ This variable was coded as 1 if migrants had heard of the services and 0 if they had not.

### 3.4. Estimation Method

The dependent variables in this study were binary indicator variables, taking values of 0 or 1. Binary logistic regression is a suitable statistical method for analyzing data with binary variables. Therefore, this model was applied to examine the relationship between the health status of migrants (0 = unhealthy, 1 = healthy) and the independent variables. The regression model is as follows:(1)$$lnP=ln\frac{exp[\beta_{0}+\sum \beta_{i}x_{i}]}{1+exp[\beta_{0}+\sum \beta_{i}x_{i}]}=\ln\left[\frac{P}{1-P}\right]=\beta_{0}+\sum \beta_{i}x_{i}$$
where $P\left(y=1|x_{1},\cdots ,x_{i}\right)$ represents the probability of health status of migrants, where $x_{i}$ denotes the demographic characteristics and other characteristic variables that influence the migrants’ health status. The values of the estimated coefficients for these variables are denoted by $\beta_{i}$.

## 4. Results

### 4.1. Descriptive Analysis

Table 1 presents the descriptive statistics for each variable. There were 169,989 migrants in the sample. On average, migrants had participated in three or more health education programs, and the average age of migrants was in their 30s. Among migrants, 51.7% were male and 84.3% were married. The educational backgrounds of the migrants were concentrated in middle and high school, and 93.2% of the migrants had health insurance. 59.9% of the migrants had heard of the National Basic Public Health Services, and 77.9% of the migrants had agricultural household registration. Reproductive health education and maternal and child health education were the two most commonly received health education categories, with more than 50% of migrants having received education in these areas. Furthermore, over 70% of migrants had received health education through publicity materials and bulletin boards.

### 4.2. Regression Results

#### 4.2.1. Effects of Health Education on Migrants’ Health Status

Table 2 presents the results of a regression analysis, in which the coefficient β estimates the change in the odds ratio of the health status reported by the respondents, as indicated by Exp. (β). An odds ratio value exceeding 1 signifies a heightened probability of good health among the respondents, whereas a value less than 1 indicates a reduced probability [51]. From the empirical results, it can be seen that health education has a positive effect on the health status of migrants (β = 0.041) at a significance level of 1%, after controlling for the demographic characteristics, socio-economic status, and health service publicity. Through this, it can be seen that receiving health education is helpful for the health of the migrants. The Cox and Snell *R*^2^, which shows the explanatory power of the logistic regression model, is estimated to be 0.054, and the Nagelkerke *R*^2^ is 0.252. Although these values are relatively low compared to the *R*^2^ in regression analysis, the assumption of equal variance of errors is not satisfied in logistic regression analysis, and the *R*^2^ varies depending on the predicted probability. Therefore, there are limitations in interpreting the explanatory power of the model based on the *R*^2^, as the *R*^2^ obtained from logistic regression analysis tends to be low [52,53]. The Hosmer–Lemeshow goodness-of-fit test is commonly used to evaluate the fit of a logistic regression model to the observed data, with small *p*-values (typically less than 0.05) indicating poor fit, and larger *p*-values closer to 1 indicating good overall fit [54]. It is important to note that Hosmer and Lemeshow have cautioned against using this test when the sample size is small, typically less than 400, as it can be overly sensitive and lead to misleading results. Given that the *p*-value obtained from the Hosmer–Lemeshow test for this model is 0.740, it can be concluded that the model fits well.

Table 3 shows the results of the impact of different types of health education. The results indicate that health education related to occupational diseases (β = 0.266), venereal disease/AIDS (β = 0.153), and self-rescue during public emergencies (β = 0.125) has a positive and significant impact on the health status of migrants. However, health education on chronic diseases (β = −0.268) is found to have a negative and significant influence on the health status of migrants.

Table 4 presents the results of the effects by educational media type. The results suggest that health education delivered through lectures (β = 0.285) and bulletin boards (β = 0.154) positively and significantly affects the health status of migrants. However, health education delivered through online platforms (β = −0.174) is found to have a negative and significant influence on the health status of migrants.

#### 4.2.2. Analysis of Heterogeneity by Gender and Age Sample

Scholars have extensively researched health inequality among migrants, with a particular focus on female migrants.These women, affected by both low socio-economic status and gender discrimination influenced by biological and socio-economic factors [55], are more vulnerable to infectious diseases such as sexually transmitted diseases [56]. These studies have suggested the effectiveness of health education for female migrants. Thus, it is necessary to further examine the impact of health education on the health status of migrants, with a focus on gender. The logistic regression results in Table 5 show that health education has a significant positive effect on the health status of both male (β = 0.031) and female (β = 0.051) migrants. These results also show that it has a greater impact on the health status of female migrants than male migrants. Therefore, it can be concluded that health education is relatively more effective in improving the health status of female migrants.

While the primary cohort of migrants comprises young adults, the trend towards family-based settlement in urban areas has resulted in an expanding demographic range among China’s migrant population [57], which now includes a growing number of middle-aged and elderly individuals [58]. This demographic shift has prompted research into the differences in disease prevalence among migrants based on age [59]. In light of this, we examined the impact of health education on the health status of migrants according to their age. Table 6 presents the effects of health education on migrants’ health status by age. The results indicate that health education has a strong significant positive impact on the health of migrants aged 40 to 59 (β = 0.034) and those aged 60 and above (β = 0.093), with a particular benefit for the health of migrants aged 60 and above.

### 4.3. Mediation Model Test

To examine the mediating pathways between health education and health status of migrants through health behaviors, this study employed structural equation modeling (SEM hereafter). SEM is a statistical method used to test complex relationships among variables. Three equations were proposed, as outlined below:(2)$$Y=cX+e_{1}$$
(3)$$M=aX+e_{2}$$
(4)$$Y=c^{\prime }X+bM+e_{3}$$

Equation (2) examines the effect of health education on the health of the migrants, where Y represents the independent variable, X is the dependent variable for health education, and the coefficient c represents the total effect of health education on the health of the migrants. Equation (3) analyzes the impact of health education on the health behaviors of migrants, where M represents the intermediary variable for health behaviors and X is the dependent variable for health education. Equation (4) explores the influence of mediating variables and independent variables on the dependent variable, where *c*′ represents the direct effect of health education on the health of migrants and *b* represents the indirect effect of mediating variables on the migrants. In each Equation, e_1_, e_2_, and e_3_ are the regression residuals. 

The mediating effect is primarily determined by sequentially testing the significance of the coefficients of the three Equations and assessing whether there is a mediating effect. This first step is to examine the coefficient *c* of Equation (2). If the coefficient *c* is significant, the mediating effect can be further analyzed. If the coefficient *c* is not significant, the generalized mediating effect can be used to determine the masking effect. The second step is to assess the significance of coefficients a and b in Equations (3) and (4). If these coefficients have significant effects, the presence of a mediating effect is indicated. If one coefficient is not significant, the bootstrap method should be used to test for the existence of a mediating effect.

The results of the empirical analysis are presented in Table 7, indicating that health education has a significant positive effect on increasing medical-seeking behavior among migrants (β = 0.072) in China, while controlling for other variables. This increase is statistically significant at the 5% level. These findings suggest that health education can enhance the health knowledge and medical-seeking behavior of migrants in China.

Meanwhile, according to Table 8, when controlling for other variables, health education significantly increases the hygiene behavior of migrants (β = 0.021) at a 0.1% significance level. This demonstrates that health education can improve health literacy and subsequently lead to changes in hygiene behavior among migrants. The first step in the mediating effect analysis is to determine the effect of health education on the health status of migrants, and the analysis results (Table 2) show that health education has a significant effect on the health status of the migrants.

The second step is to determine whether health education has a significant effect on the health behaviors, and the results in Table 7 and Table 8 show that health education had a significant positive effect on the health behaviors of the migrants. Finally, health education and health behaviors are included in the equation at the same time to determine the mediating effect through their significance, and the empirical results in Table 9 show that health education affects the health status of the migrants (β = 0.050) through their medical-seeking behavior (β = 0.956). With the medical-seeking behavior unchanged, the effect of health education is mainly due to the improvement of health knowledge, and health education has a significant positive effect on the health of the migrants, indicating that migrants with more health knowledge are healthier.

Meanwhile, Table 10 presents the empirical analysis of the mechanism by which health education affects the health status of migrants through their hygiene behavior. The analysis shows that health education does not have a significant effect on the health status of migrants (β = 0.041) through their hygiene behavior (β = 0.013). However, it needs to be further verified whether health education affects the hygiene behavior of migrants and subsequently affects their health status. Although hygiene behavior does not significantly affect the health status of migrants when both variables are included in the analysis, additional tests are required to obtain accurate results. This is because health education has a significant effect on the hygiene behavior of migrants in China.

This study employed SEM with a maximum likelihood estimation and a bootstrap test to examine the mediation effect of medical-seeking behavior and hygiene behavior. The bootstrap test was applied using 169,989 samples to test the 95% bias-corrected confidence interval for indirect effects. The data analysis was conducted using IBM SPSS Amos version 29. The results of the tests, presented in Table 11, show that the indirect effects of each mediating variable, such as medical-seeking behavior and hygiene behavior, were not significant, while the total effect was significant. This indicates that medical-seeking behavior and hygiene behavior can act as mediating factors between health education and the health status of migrants. In summary, the results suggest that health education can influence the health status of migrants by modifying their medical-seeking and hygiene behaviors.

## 5. Discussion

Health education has a significant positive impact on the health status of migrants in China. Health education on occupational diseases, venereal diseases/AIDS, and self-rescue in public emergencies has a significant positive impact on the health status of migrants. Due to the high risks associated with occupational diseases and AIDS among migrants in China, previous studies on migrant health in China have predominantly focused on addressing these specific issues [60,61]. Based on our findings, it is evident that providing public health education on occupational diseases and AIDS can effectively address the health needs of migrants in China. Meanwhile, health education on chronic diseases has a significant negative impact on migrants’ health status, indicating that chronic disease education is not efficient. However, there is a lack of research on chronic diseases among migrants in China [62]. In terms of chronic disease health education, migrants receive the lowest coverage in chronic disease management services in China [63]. Chronic diseases pose long-term health risks, and the positive impact of interventions focusing on chronic disease health education has already been established in previous studies [64,65]. Therefore, efficient education on chronic diseases for the migrants in China could help reduce the risk of chronic diseases that arise from socio-economic disadvantages, thereby contributing to mitigating the chronic disease burden among migrants in China.

In terms of educational media, traditional media such as lectures and bulletin boards have a significant positive impact on the health status of migrants, while online education has a significant negative impact. Considering the effectiveness of face-to-face education as indicated by previous studies [66,67,68], the positive effects of lectures become evident. However, there is a study showing that low health literacy and poor health status is also related to the limited utilization of online health education [69]. Therefore, there is room for online health education to develop. The effect of online education on the health intervention of the migrant population needs to be further discussed.

The regression analysis of gender and age for the sample of the migrant population shows that there are differences in the effectiveness of health education among different genders and ages of the migrant population. In terms of gender, both male and female migrants show significant results, but the impact of health education on the health of female migrants is notably more significant. As mentioned in Section 4, the overall health status of female migrants tends to be lower than that of male migrants. Therefore, when addressing the health needs of the migrants in China, the quality of services should be adjusted to account for the quality of life of female migrants. As for age, health education is more likely to yield positive health status for older migrants. This finding indicates that, on the one hand, public health education can improve the health status of older migrants. On the other hand, it may not be as effective in improving the health of younger migrants, particularly those in their 30s. Targeted public health education for younger migrants can help identify and prevent diseases at an early stage. As mentioned in Section 4, despite the need to analyze and differentiate the health effects of public health education on the migrants based on gender and age, considering the socio-economic status of migrants and migration trends, there is a scarcity of relevant research. Therefore, more studies addressing the health effects considering the gender and age of migrants should be conducted.

Through the mediating test analysis, it was observed that health behaviors had a mediating effect between health education and the health status of migrants, as confirmed by the bootstrap test. Health education affects the health status of migrants by influencing their health behaviors. Health education has a significant positive impact on health behaviors, and these behaviors, in turn, have a positive effect on the health status of migrants. Therefore, modifying the health behaviors of migrants can lead to a better health status. These research findings are consistent with the mechanisms proposed by health behavior theories, such as the HBM, which suggests that health knowledge enhances health behaviors and overall health status. However, further extensive research on the specific mechanisms applicable to the migrant population in China is needed to provide additional support for our study findings. 

This study suggests several policy suggestions. Firstly, to enhance the quality of community-based public health education programs, the Chinese government should encourage their development and innovation. It is essential to develop a comprehensive public health knowledge system based on health behavior theories for universities, students in relevant departments, and practitioners in the field. Secondly, the findings of this study suggest that the Chinese government should implement measures to address non-communicable diseases, placing particular emphasis on chronic diseases in addition to infectious diseases. Based on the on-site survey conducted in Jiangsu Province in May 2023, it is evident that education on chronic diseases remains a significant challenge that the Chinese government still needs to address. Thirdly, in public health education, prioritizing the health status of female migrants and elderly migrants is essential due to their heightened responsiveness to such initiatives. Conversely, targeting young migrants with public health education becomes crucial for early disease prevention. Lastly, traditional media should be utilized more for health education due to its high accessibility and minimal technical issues [70]. It is worth noting that despite the potential benefits, online health education has demonstrated a negative impact on the health of migrants in China. This can be attributed to technical issues and low levels of accessibility and utilization. Therefore, both the Chinese government and communities should prioritize increasing the accessibility of Internet-based public health education programs for migrants in China.

This study possesses strengths in several aspects. While previous studies have examined the relationship between health education and the health status of migrants in China, limited attention has been given to understanding the mechanisms through which health education leads to improved health status via changes in health behaviors. Additionally, there is a scarcity of quantitative model exploration regarding the impact of health education on the health status of migrants in China. This study addresses these research gaps by conducting quantitative model exploration to shed light on the effect of health education on the health status of migrants. Furthermore, although effective communication is crucial in health education, determining the most effective method for migrants in China, considering their diverse life experiences, remains uncertain. This study contributes to bridging this research gap by investigating and identifying effective communication methods tailored for migrants in China.

This study has several limitations in the following aspects. Firstly, the effect of health education on the health status of the migrant population was measured solely through SRHS, which is a comprehensive health indicator. Future studies can consider examining the impact of health education on various dimensions of health, such as physical, psychological, social, and environmental aspects [71,72]. Secondly, this study primarily focused on the individual-level impact of health education on migrants in China. Future studies can gather information on health education for migrants from a health environment perspective (i.e., the meso and macro levels) and explore how communities and governments can utilize different empirical research strategies to balance cost-effectiveness with the health needs of migrants in China. Thirdly, while this study conducted heterogeneity analysis of the effects of health education on the health status of migrants based on gender and age, future research could explore a broader range of socio-economic factors among the migrants in China, considering that additional socio-economic factors would provide valuable insights into how health education can effectively target the specific needs of migrants in China.

## 6. Conclusions

This study examined the influence of health education on the health status of migrants in China. The findings revealed a significant positive effect of health education on the health status of migrants in China. Specifically, education on occupational diseases, venereal diseases/AIDS, and self-rescue in public emergencies exhibited a significant positive impact on the health status of migrants in China, while education on chronic diseases had a significant negative impact. Lectures and bulletin boards had a significant positive impact on the health status of migrants in China, while online education had a significant negative impact. Notably, the impact of health education on the health status of female migrants and elderly migrants was found to be more significant. It was observed that health education can enhance the health status of migrants by influencing their medical-seeking behavior and hygiene behavior.

Enhancing the health of migrants has significant implications for China’s economic development and societal stability. There are several advantages to strengthening health education for this population. Public health education is recognized as a highly cost-effective intervention that can yield substantial health improvements. In order to maximize its impact, China’s public health education efforts should employ persuasive and effective means that are tailored to the specific health behaviors and needs of migrants, taking into account their socio-economic status. By implementing differentiated approaches, public health education programs can effectively drive positive changes in the health behavior of migrants and contribute to their overall well-being.

## Figures and Tables

**Figure 1 healthcare-11-01768-f001:**
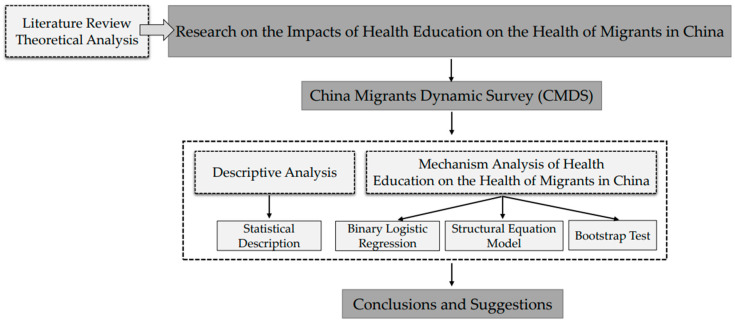
Analytical framework.

**Figure 2 healthcare-11-01768-f002:**
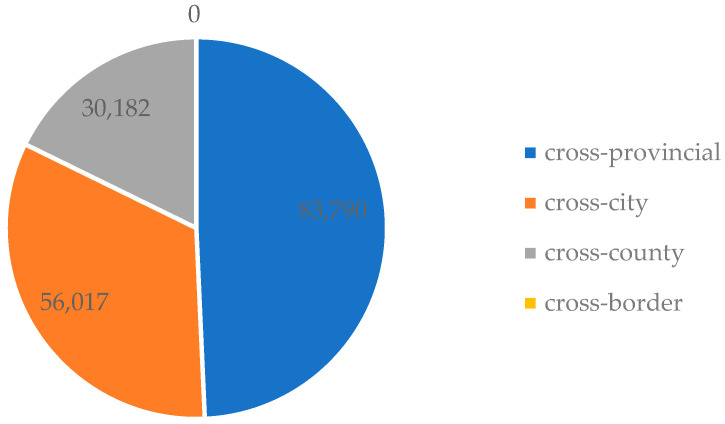
Flowchart of China’s migrant population (unit: individuals).

**Table 1 healthcare-11-01768-t001:** Statistical description.

Variable	Observation	Mean	Standard Deviation
**Dependent variables**			
Health status	169,989	0.972	0.162
**Independent variables**			
Health education	154,586	3.753	3.388
Occupational disease	154,586	0.333	0.471
Venereal disease/AIDS	154,586	0.396	0.489
Reproductive health	154,586	0.504	0.499
Mental health	154,586	0.357	0.479
Chronic disease	154,586	0.374	0.483
Maternal and child health	154,586	0.511	0.499
Self-rescue in public emergency	154,586	0.422	0.493
Lecture	112,987	0.446	0.497
Publicity material	112,987	0.856	0.351
Bulletin board	112,987	0.748	0.434
Public consultation	112,987	0.453	0.498
Online education	112,987	0.301	0.458
**Mediation variables**			
Medical-seeking behavior	41,287	0.993	0.077
Hygiene behavior	169,989	0.801	0.399
**Control variables**			
Gender	169,989	0.516	0.500
Age	166,695	2.253	1.110
Education level	152,210	2.209	0.837
Household registration	169,989	0.779	0.414
Monthly household income	169,982	2.362	1.127
Employment status	169,989	0.822	0.382
Social security card	159,525	0.533	0.499
Health service publicity	169,989	0.599	0.490
Basic medical insurance	167,034	0.932	0.250
Marriage	163,769	0.843	0.364
Migration range	139,807	1.599	0.490
Current residence	162,990	1.803	0.820

**Table 2 healthcare-11-01768-t002:** Logistic regression results of health education and control variables on the health status of migrants.

Variables	Health Status		
	β (S.E.)	Wald	Exp. (β)
Health education	0.041 *** (0.008)	29.245	1.042
Gender_male	0.105 * (0.046)	5.164	1.110
Age_30–39	−0.969 *** (0.125)	60.417	0.380
Age_40–49	−1.987 *** (0.120)	276.119	0.137
Age_50–59	−2.596 *** (0.122)	453.477	0.075
Age_60–69	−2.815 *** (0.130)	466.677	0.060
Age_70 or older	−3.228 *** (0.155)	435.298	0.040
Education level_middle school	0.643 *** (0.050)	165.487	1.902
Education level_high school	0.860 *** (0.076)	128.004	2.363
Education level_college and above	1.395 *** (0.206)	45.755	4.036
Household registration_agricultural household registration	−0.017 (0.063)	0.070	0.984
Household income_CNY 4000–6000	0.494 *** (0.053)	87.995	1.639
Household income_CNY 6001–8000	0.786 *** (0.075)	110.571	2.195
Household income_CNY more than 8001	1.049 *** (0.075)	193.893	2.855
Employment status_yes	1.333 *** (0.048)	757.579	3.791
Social security card_yes	−0.009 (0.046)	0.039	0.991
Health service publicity_yes	0.277 *** (0.047)	34.370	1.319
Basic medical insurance_yes	0.046 (0.078)	0.339	1.047
Marriage_yes	−0.111 (0.133)	0.699	0.895
Migration range_across province	0.120 ** (0.047)	6.470	1.128
Current residence_middle	−0.361 *** (0.059)	37.513	0.697
Current residence_west	−0.395 *** (0.056)	49.388	0.674
Constant	3.379 *** (0.179)	356.390	29.341
−2 Log likelihood = 17,644.321			
Model *χ*^2^ = 5263.634 ***			
Cox and Snell *R*² = 0.054			
Nagelkerke *R*² = 0.252			
Hosmer and Lemeshow = 5.159 (*p*-value = 0.740)			
Observation = 94,517			

Note: *** *p* < 0.001, ** *p* < 0.01, * *p* < 0.05.

**Table 3 healthcare-11-01768-t003:** Logistic regression results of health education and control variables on the health status of migrants (by education type).

Variables	Health Status		
	β (S.E.)	Wald	Exp. (β)
Occupational disease	0.266 *** (0.072)	13.758	1.305
Venereal disease/AIDS	0.153 * (0.072)	4.462	1.165
Reproductive health	0.022 (0.070)	0.101	1.022
Mental health	0.025 (0.069)	0.128	1.025
Chronic disease	−0.268 *** (0.066)	16.459	0.765
Maternal child health	0.111 (0.066)	2.814	1.117
Self-rescue in public emergency	0.125 * (0.062)	4.111	1.133
Gender_male	0.105 * (0.046)	5.108	1.111
Age_30–39	−0.964 *** (0.125)	59.756	0.381
Age_40–49	−1.971 *** (0.120)	270.627	0.139
Age_50–59	−2.565 *** (0.123)	436.803	0.077
Age_60–69	−2.758 *** (0.132)	436.653	0.063
Age_70 or older	−3.168 *** (0.156)	410.497	0.042
Education level_middle school	0.636 *** (0.050)	161.349	1.888
Education level_high school	0.851 *** (0.076)	125.223	2.343
Education level_college and above	1.380 *** (0.206)	44.717	3.976
Household registration_agricultural household registration	−0.024 (0.063)	0.143	0.977
Household income_CNY 4000–6000	0.490 *** (0.053)	86.551	1.633
Household income_CNY 6001–8000	0.778 *** (0.075)	108.069	2.177
Household income_more than CNY 8001	1.043 *** (0.075)	191.485	2.839
Employment status	1.317 *** (0.049)	734.136	3.734
Social security card_yes	−0.008 (0.046)	0.030	0.992
Health service publicity_yes	0.287 *** (0.047)	36.668	1.332
Basic medical insurance_yes	0.048 (0.078)	0.369	1.049
Marriage_yes	−0.111 (0.134)	0.692	0.895
Migration range_across province	0.115 * (0.047)	5.868	1.121
Current residence_the central	−0.352 *** (0.059)	35.472	0.703
Current residence_the west	−0.389 *** (0.056)	47.546	0.678
Constant	3.373 *** (0.179)	354.139	29.171
−2 Log likelihood = 17,608.477			
Model *χ*^2^ = 5299.478 ***			
Cox and Snell *R*² = 0.055			
Nagelkerke *R*² = 0.253			
Hosmer and Lemeshow = 4.113 (*p*-value = 0.847)			
Observation = 94,517			

Note: *** *p* < 0.001, * *p* < 0.05.

**Table 4 healthcare-11-01768-t004:** Logistic regression results of education media and control variables on the health status of migrants.

Variables	Health Status		
	β (S.E.)	Wald	Exp. (β)
Lecture	0.285 *** (0.065)	19.138	1.330
Publicity material	0.136 (0.074)	3.383	1.146
Bulletin board	0.154 * (0.065)	5.661	1.167
Public consultation	0.042 (0.069)	0.371	1.043
Online education	−0.174 * (0.069)	6.428	0.841
Gender_male	0.105 (0.059)	3.177	1.111
Age_30–39	−0.939 *** (0.154)	37.167	0.391
Age_40–49	−2.008 *** (0.148)	184.422	0.134
Age_50–59	−2.644 *** (0.152)	303.511	0.071
Age_60–69	−2.774 *** (0.166)	280.192	0.062
Age_70 or older	−3.236 *** (0.201)	259.698	0.039
Education level_middle school	0.647 *** (0.063)	104.293	1.910
Education level_high school	0.896 *** (0.097)	84.755	2.450
Education level_college and above	1.615 *** (0.293)	30.464	5.029
Household registration_agricultural household registration	0.018 (0.079)	0.054	1.019
Household income_CNY 4000–6000	0.601 *** (0.067)	79.424	1.823
Household income_CNY 6001–8000	0.778 *** (0.092)	70.777	2.177
Household income_more than CNY 8001	1.133 *** (0.099)	131.062	3.103
Employment status	1.310 *** (0.062)	450.634	3.707
Social security card_yes	−0.035 (0.058)	0.356	0.966
Health service publicity_yes	0.286 *** (0.060)	22.776	1.331
Basic medical insurance_yes	−0.022 (0.107)	0.044	0.978
Marriage_yes	−0.342 (0.187)	3.355	0.710
Migration range_across province	0.117 * (0.059)	3.893	1.124
Current residence_the central	−0.342 *** (0.077)	19.500	0.710
Current residence_the west	−0.390 *** (0.072)	29.673	0.677
Constant	3.529 *** (0.250)	199.892	34.099
−2 Log likelihood = 11,121.878			
Model *χ*^2^ = 3174.642 ***			
Cox and Snell *R*² = 0.045			
Nagelkerke *R*² = 0.240			
Hosmer and Lemeshow = 8.753 (*p* value = 0.364)			
Observation = 77,156			

Note: *** *p* < 0.001, * *p* < 0.05.

**Table 5 healthcare-11-01768-t005:** Logistic regression results of health education on the health status of migrants by gender.

Variables	Male			Female		
	β (S.E.)	Wald	Exp. (β)	β (S.E.)	Wald	Exp. (β)
Health education	0.031 ** (0.011)	8.427	1.032	0.051 *** (0.011)	22.498	1.052
Age_30–39	−0.662 *** (0.201)	10.848	0.516	−1.093 *** (0.161)	46.184	0.335
Age_40–49	−1.704 *** (0.191)	79.254	0.182	−2.049 *** (0.156)	172.786	0.129
Age_50–59	−2.282 *** (0.194)	138.095	0.102	−2.691 *** (0.159)	284.774	0.068
Age_60–69	−2.487 *** (0.206)	145.976	0.083	−2.898 *** (0.172)	283.568	0.055
Age_70 or older	−2.921 *** (0.228)	163.501	0.054	−3.189 *** (0.227)	197.641	0.041
Education level_middle school	0.508 *** (0.072)	49.998	1.662	0.762 *** (0.071)	115.497	2.142
Education level_high school	0.697 *** (0.103)	46.174	2.009	1.021 *** (0.116)	78.080	2.776
Education level_college and above	1.281 *** (0.280)	20.980	3.600	1.512 *** (0.308)	24.192	4.538
Household registration_agricultural household registration	−0.075 (0.090)	0.692	0.928	0.033 (0.087)	0.139	1.033
Household income_CNY 4001–6000	0.446 *** (0.077)	33.995	1.562	0.528 *** (0.073)	52.739	1.696
Household income_CNY 6001–8000	0.820 *** (0.111)	54.693	2.271	0.758 *** (0.101)	55.837	2.135
Household income_more than CNY 8001	1.032 *** (0.111)	86.615	2.806	1.058 *** (0.103)	105.743	2.881
Employment status_yes	1.577 *** (0.073)	469.649	4.841	1.149 *** (0.064)	319.835	3.154
Social security card_yes	0.026 (0.067)	0.153	1.026	−0.036 (0.063)	0.320	0.965
Health service publicity_yes	0.293 *** (0.068)	18.300	1.340	0.254 *** (0.065)	15.133	1.289
Basic medical insurance_yes	−0.004 (0.118)	0.001	0.996	0.081 (0.105)	0.590	1.084
Marriage_yes	−0.150 (0.166)	0.820	0.860	−0.199 ** (0.239)	0.697	0.819
Migration range_across province	0.128 (0.069)	3.465	1.136	0.119 (0.065)	3.312	1.126
Current residence_the central	−0.379 *** (0.086)	19.435	0.685	−0.347 *** (0.081)	18.182	0.707
Current residence_the west	−0.335 *** (0.082)	16.543	0.716	−0.437 *** (0.077)	32.112	0.646
Constant	3.250 *** (0.248)	172.135	25.780	3.493 *** (0.283)	152.829	32.883
−2 Log likelihood	8576.983			9028.597		
Model *χ*^2^	2461.923 ***			2802.214 ***		
Cox and Snell *R*²	0.048			0.061		
Nagelkerke *R*²	0.242			0.261		
Hosmer and Lemeshow	5.720 (*p*-value = 0.679)			9.949 (*p*-value = 0.269)		
Observation	50,149			44,368		

Note: *** *p* < 0.001, ** *p* < 0.01.

**Table 6 healthcare-11-01768-t006:** Logistic regression results of health education on health status of migrants by age.

Variables	20 to 29		30 to 39		40 to 59		60 and above	
	β (S.E.)	Exp. (β)	β (S.E.)	Exp. (β)	β (S.E.)	Exp. (β)	β (S.E.)	Exp. (β)
Health education	0.036 (0.036)	1.037	0.029 (0.020)	1.030	0.034 *** (0.010)	1.035	0.093 *** (0.019)	1.097
Gender_male	−0.137 (0.238)	0.872	0.207 (0.127)	1.230	0.102 (0.058)	1.107	0.028 (0.102)	1.028
Middle school	1.708 *** (0.266)	5.517	1.083 *** (0.133)	2.953	0.611 *** (0.061)	1.843	0.467 *** (0.121)	1.596
High school	1.749 *** (0.305)	5.751	1.652 *** (0.203)	5.216	0.743 *** (0.100)	2.102	0.548 *** (0.163)	1.731
College and above	1.661 *** (0.483)	5.266	1.987 *** (0.366)	7.292	2.089 *** (0.587)	8.078	0.749 (0.396)	2.115
Agricultural household registration	−1.856 * (0.730)	0.156	0.161 (0.179)	1.174	0.088 (0.082)	1.092	−0.131 (0.124)	0.878
Income_CNY 4001–6000	0.300 (0.261)	1.349	0.596 *** (0.141)	1.814	0.524 *** (0.066)	1.689	0.420 ** (0.129)	1.522
Income_CNY 6001–8000	0.612 (0.385)	1.844	0.835 *** (0.183)	2.305	0.789 *** (0.092)	2.202	0.991 *** (0.208)	2.694
Income_more than CNY 8001	0.213 (0.320)	1.237	1.139 *** (0.201)	3.122	1.179 *** (0.101)	3.251	0.850 *** (0.159)	2.339
Employment status_yes	1.112 *** (0.236)	3.041	0.958 *** (0.131)	2.607	1.532 *** (0.058)	4.629	1.244 *** (0.126)	3.471
Social security card_yes	0.162 (0.226)	1.176	−0.166 (0.123)	0.847	−0.035 (0.058)	0.966	0.133 (0.107)	1.142
Health service publicity_yes	0.131 (0.231)	1.140	0.656 *** (0.127)	1.928	0.275 *** (0.060)	1.317	0.064 (0.104)	1.067
Basic medical insurance_yes	−0.901 (0.521)	0.406	0.315 (0.190)	1.371	0.070 (0.098)	1.073	−0.193 (0.188)	0.825
Marriage_yes	0.285 (0.249)	1.329	−0.749 * (0.346)	0.473	−0.048 (0.206)	0.953	−1.184 (0.759)	0.306
Migration range_across province	−0.149 (0.232)	0.861	0.124 (0.125)	1.132	0.177 ** (0.060)	1.193	−0.029 (0.108)	0.972
Current residence_the central	−0.263 (0.320)	0.769	−0.376 * (0.156)	0.686	−0.281 *** (0.075)	0.755	−0.667 *** (0.133)	0.513
Current residence_the west	−0.611 * (0.270)	0.543	−0.159 (0.153)	0.853	−0.355 *** (0.070)	0.701	−0.607 *** (0.135)	0.545
Constant	5.635 *** (0.947)	280.053	2.267 *** (0.428)	9.649	0.835 *** (0.238)	2.304	2.174 ** (0.782)	8.794
−2 Log likelihood	1072.090		3098.415		10,696.870		2754.352	
Model *χ^2^*	124.053 ***		384.129 ***		1585.087 ***		345.006 ***	
Cox and Snell *R²*	0.005		0.012		0.045		0.097	
Nagelkerke *R²*	0.106		0.116		0.150		0.162	
Hosmer and Lemeshow	2.738 (*p*-value = 0.950)		10.264 (*p*-value = 0.247)		12.924 (*p*-value = 0.114)		8.959 (*p*-value = 0.346)	
Observation	23,980		32,972		34,192		3373	

Note: *** *p* < 0.001, ** *p* < 0.01, * *p* < 0.05.

**Table 7 healthcare-11-01768-t007:** Analysis of the impact of health education on the medical-seeking behavior of migrants.

Variables	Medical-Seeking Behavior		
	β (S.E.)	Wald	Exp. (β)
Health education	0.072 * (0.030)	5.597	1.074
Gender_male	−0.152 (0.181)	0.699	0.859
Age_30–39	−0.150 (0.285)	0.276	0.861
Age_40–49	−0.620 * (0.296)	4.385	0.538
Age_50–59	−0.755 * (0.346)	4.776	0.470
Age_60–69	−0.970 * (0.426)	5.189	0.379
Age_70 or older	−0.219 (0.789)	0.077	0.803
Education level_middle school	0.311 (0.221)	1.975	1.365
Education level_high school	0.333 (0.280)	1.416	1.395
Education level_college and above	0.696 (0.454)	2.353	2.006
Household registration_agricultural household registration	0.734 *** (0.212)	12.028	2.083
Household income_CNY 4000–6000	0.300 (0.220)	1.868	1.350
Household income_CNY 6001–8000	0.531 (0.280)	3.598	1.700
Household income_more than CNY 8001	0.600 * (0.264)	5.176	1.822
Employment status_yes	0.144 (0.227)	0.400	1.154
Social security card_yes	0.010 (0.186)	0.003	1.010
Health service publicity_yes	0.221 (0.185)	1.428	1.247
Basic medical insurance_yes	0.707 ** (0.269)	6.936	2.029
Marriage_yes	0.378 (0.318)	1.406	1.459
Migration range_across province	0.158 (0.189)	0.701	1.171
Current residence_middle	−0.400 (0.223)	3.213	0.670
Current residence_west	0.173 (0.233)	0.549	1.189
Constant	2.896 *** (0.492)	34.692	18.097
−2 Log likelihood = 1608.801			
Model *χ*^2^ = 79.345 ***			
Cox and Snell *R*² = 0.003			
Nagelkerke *R*² = 0.049			
Hosmer and Lemeshow = 9.541 (*p*-value = 0.299)			
Observation = 23,080			

Note: *** *p* < 0.001, ** *p* < 0.01, * *p* < 0.05.

**Table 8 healthcare-11-01768-t008:** Analysis of the impact of health education on the hygiene behavior of migrants.

Variables	Hygiene Behavior		
	β (S.E.)	Wald	Exp. (β)
Health education	0.021 *** (0.003)	59.708	1.021
Gender_male	−0.045 ** (0.017)	6.949	0.956
Age_30–39	−0.005 (0.024)	0.051	0.995
Age_40–49	−0.089 *** (0.026)	11.761	0.915
Age_50–59	−0.116 *** (0.033)	12.433	0.890
Age_60–69	0.022 (0.053)	0.179	1.022
Age_70 or older	0.147 (0.104)	2.009	1.158
Education level_middle school	0.292 *** (0.022)	183.220	1.340
Education level_high school	0.543 *** (0.027)	397.356	1.721
Education level_college and above	0.983 *** (0.046)	465.012	2.673
Household registration_agricultural household registration	−0.113 *** (0.024)	21.795	0.893
Household income_CNY 4000–6000	0.074 *** (0.022)	11.542	1.077
Household income_CNY 6001–8000	0.202 *** (0.026)	61.809	1.224
Household income_more than CNY 8001	0.268 *** (0.025)	114.425	1.307
Employment status_yes	0.071 ** (0.023)	9.333	1.074
Social security card_yes	0.071 *** (0.017)	17.192	1.074
Health service publicity_yes	0.140 *** (0.018)	61.860	1.150
Basic medical insurance_yes	0.023 (0.032)	0.522	1.023
Marriage_yes	0.060 (0.031)	3.803	1.062
Migration range_across province	−0.342 *** (0.018)	346.788	0.710
Current residence_middle	0.060 ** (0.023)	6.899	1.062
Current residence_west	−0.337 *** (0.020)	274.326	0.714
Constant	0.998 *** (0.053)	347.968	2.712
−2 Log likelihood = 93,839.969			
Model *χ*^2^ = 2968.236 ***			
Cox and Snell *R*² = 0.031			
Nagelkerke *R*² = 0.048			
Hosmer and Lemeshow = 12.363 (*p*-value = 0.136)			
Observation = 94,517			

Note: *** *p* < 0.001, ** *p* < 0.01.

**Table 9 healthcare-11-01768-t009:** The mechanism of the effect of health education on the health level of migrants (path validation of medical-seeking behavior).

Variables	Health Status		
	β (S.E.)	Wald	Exp. (β)
Health education	0.050 *** (0.012)	16.051	1.051
Medical-seeking behavior	0.956 *** (0.273)	12.260	2.602
Gender_male	0.168 * (0.076)	4.949	1.183
Age_30–39	−1.122 *** (0.198)	31.953	0.326
Age_40–49	−2.106 *** (0.193)	119.382	0.122
Age_50–59	−2.806 *** (0.197)	202.357	0.060
Age_60–69	−3.040 *** (0.214)	201.278	0.048
Age_70 or older	−3.573 *** (0.254)	198.315	0.028
Education level_middle school	0.675 *** (0.082)	68.397	1.964
Education level_high school	1.038 *** (0.134)	59.604	2.822
Education level_college and above	1.157 *** (0.316)	13.453	3.182
Household registration_agricultural household registration	−0.086 (0.109)	0.618	0.918
Household income_CNY 4000–6000	0.435 *** (0.085)	25.886	1.545
Household income_CNY 6001–8000	0.915 *** (0.124)	54.655	2.497
Household income_more than CNY 8001	1.189 *** (0.121)	97.342	3.284
Employment status_yes	1.159 *** (0.079)	213.317	3.186
Social security card_yes	0.040 (0.075)	0.282	1.041
Health service publicity_yes	0.296 *** (0.077)	14.655	1.345
Basic medical insurance_yes	0.030 (0.139)	0.045	1.030
Marriage_yes	−0.339 (0.236)	2.067	0.712
Migration range_across province	0.022 (0.079)	0.081	1.023
Current residence_middle	−0.466 *** (0.101)	21.268	0.628
Current residence_west	−0.272 ** (0.091)	8.997	0.762
Constant	2.261 *** (0.406)	31.071	9.594
−2 Log likelihood = 6011.141			
Model *χ*^2^ = 2241.384 ***			
Cox and Snell *R*² = 0.093			
Nagelkerke *R*² = 0.308			
Hosmer and Lemeshow = 7.122 (*p*-value = 0.524)			
Observation = 23,080			

Note: *** *p* < 0.001, ** *p* < 0.01, * *p* < 0.05.

**Table 10 healthcare-11-01768-t010:** The mechanism of the effect of health education on the health status of migrants (path validation of hygiene behavior).

Variables	Health Status		
	β (S.E.)	Wald	Exp. (β)
Health education	0.041 *** (0.008)	29.160	1.042
Hygiene behavior	0.013 (0.050)	0.064	1.013
Gender_male	0.105 * (0.046)	5.183	1.111
Age_30–39	−0.968 *** (0.125)	60.365	0.380
Age_40–49	−1.986 *** (0.120)	275.901	0.137
Age_50–59	−2.595 *** (0.122)	453.199	0.075
Age_60–69	−2.814 *** (0.130)	466.623	0.060
Age_70 or older	−3.227 *** (0.155)	435.310	0.040
Education level_middle school	0.642 *** (0.050)	164.930	1.901
Education level_high school	0.859 *** (0.076)	127.460	2.361
Education level_college and above	1.394 *** (0.206)	45.608	4.030
Household registration_agricultural household registration	−0.016 (0.063)	0.068	0.984
Household income_CNY 4000–6000	0.494 *** (0.053)	87.856	1.638
Household income_CNY 6001–8000	0.786 *** (0.075)	110.382	2.194
Household income_more than CNY 8001	1.048 *** (0.075)	193.469	2.853
Employment status_yes	1.332 *** (0.048)	757.369	3.790
Social security card_yes	−0.009 (0.046)	0.042	0.991
Health service publicity_yes	0.276 *** (0.047)	34.222	1.318
Basic medical insurance_yes	0.045 (0.078)	0.335	1.046
Marriage_yes	−0.112 (0.133)	0.710	0.894
Migration range_across province	0.121 * (0.047)	6.519	1.128
Current residence_middle	−0.362 *** (0.059)	37.548	0.697
Current residence_west	−0.394 *** (0.056)	49.150	0.674
Constant	3.371 *** (0.182)	342.900	29.097
−2 Log likelihood = 17,644.257			
Model *χ*^2^ = 5263.697 ***			
Cox and Snell *R*² = 0.054			
Nagelkerke *R*² = 0.252			
Hosmer and Lemeshow = 6.112 (*p*-value = 0.635)			
Observation = 94,517			

Note: *** *p* < 0.001, * *p* < 0.05.

**Table 11 healthcare-11-01768-t011:** Verification of the mediating effect of health education on the health status of migrants.

	Effect	β	S.E.	C.I
Health education → hygiene behavior	Direct	0.055 **	0.002	0.050~0.059
	Indirect	0.000	0.000	0.000~0.000
	Total	0.055 **	0.002	0.050~0.059
Health education → health status	Direct	0.047 **	0.002	0.042~0.052
	Indirect	0.005 **	0.000	0.004~0.005
	Total	0.052 **	0.002	0.047~0.056
Hygiene behavior → health status	Direct	0.022 **	0.003	0.016~0.027
	Indirect	0.000	0.000	0.000~0.000
	Total	0.009 **	0.003	0.016~0.027
Health education → medical-seeking behavior	Direct	0.048 **	0.002	0.043~0.053
	Indirect	0.000	0.000	0.000~0.000
	Total	0.048 **	0.002	0.043~0.053
Health education → health status	Direct	0.047 **	0.002	0.042~0.052
	Indirect	0.005 **	0.000	0.004~0.005
	Total	0.052 **	0.002	0.047~0.056
Medical-seeking behavior → health status	Direct	0.072 **	0.006	0.062~0.084
	Indirect	0.000	0.000	0.000~0.000
	Total	0.073 **	0.006	0.062~0.083

Note: ** *p* < 0.01.

## Data Availability

Dataset available via the Migrant Population Service Center, National Health Commission, China.

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
