# Peer review of "Relationships between Health Education, Health Behaviors, and Health Status among Migrants in China: A Cross-Sectional Study Based on the China Migrant Dynamic Survey"

_healthcare, 2023, doi:10.3390/healthcare11121768_

Round 1

Reviewer 1 Report

Attached

Author Response

Response to Reviewer 1 Comments

Point 1: Try to avoid using abbreviation in abstract like CMDS?

Response 1: In the second sentence of the abstract, we explicitly mentioned the original name, ‘China Migrants Dynamic Survey’, instead of using the abbreviation CMDS.

Point 2: The authors should introduce the cencepts of “Health Behaviors, and Health Outcomes” similarly as they have done for “Health Education”.

Response 2: (p. 2) Drawing on relevant literature in the field, we expounded upon the definitions of health behaviors and health status in the introduction section. Upon careful review of the manuscript, it was determined to replace the term “health outcomes” with “health status”. This decision was based on the fact that the health variable used in our analysis was particularly related to self-rated health status (SRHS). Therefore, we would like to inform you of our reconsideration of the terminology selection.

                We offered the following elucidations for the two concepts introduced in the introduction section. Health behaviors refer to “personal attributes such as beliefs, expectations, motives, values, perceptions, and other cognitive elements; personality characteristics, including affective and emotional states and traits; and overt behavior patterns, actions, and habits that relate to health maintenance, to health restoration and to health improvement” (Gochman, 1989, p. 169). Health status is commonly defined as individuals’ self-perceived health (Centers for Disease Control and Prevention, 2022). This multidimensional concept encompasses various aspects such as physical, cognitive, emotional, and social well-being, as well as the presence or absence of disabilities (Stewart, Ware, and Ware Jr, 1992). Health surveys often capture these dimensions to assess an individual’s overall health status. It is often referred to as self-rated health status and has consistently shown its significance in predicting important health outcomes including mortality and morbidity (Dowd and Zajacova, 2007).

Point 3: The research design used for this study is appropriate. The use of multistage random sampling techniques is very suitable for this type of study. But to select a participant (to reach a sample of 169,989), did you use random number tables and random number generator software or any other technique, needs clarification? If I understand correctly, these data were drawn from the China Migrants Dynamics Survey (CMDS) series of comprehensive migration censuses conducted by the National Health Commission of China between 2009 and 2018. Could you specify which specific series/year data was/were used? Did you use all the data between 2009 & 2018, if so, state so?

Response 3: (p. 4-5) We conducted quantitative analysis utilizing the CMDS 2017 database, which contains survey items related to public health education. A total of 169,989 samples were extracted from the CMDS 2017 database. IBM SPSS Statistics version 29 was employed as the analysis program, which functions as a random number generator software. In Section 3.1, Data Source, we indicated the analysis program and the year of the database used.

Point 4: A statement with regards to ethical approval of the research is required. Was there an ethical waiver?

Response 4: (p. 25) The use of China Migrants Dynamic Survey 2017 data did not require any additional ethical approval, as ethical approval was already granted by the Ethics Review Board of the National Health Commission for the China Migrants Dynamic Survey 2017 data. Written informed consent was obtained from all participants as well. Since our study involved the analysis of de-identified existing data, no further ethical approval was necessary. We have included an Institutional Review Board Statement section in the final part of our manuscript, where we have provided a statement regarding the ethical approval of the research.

Point 5: The discussion section should be strengthened. Besides citing relevant literature, all findings under main variables should be discussed with convincing explanations.

Response 5: (p. 23-25) Based on the primary findings, we have significantly augmented the content of the discussion section. This section now encompasses the establishment of associations with previous studies, the exploration of policy implications, and the strengths and limitations of the current study. The revised content includes the following aspects. For a more detailed discussion, please refer to our manuscript.

                We discussed our research findings in relation to previous studies in the field. Firstly, health education has a significant positive impact on the health status of migrants in China, particularly in relation to occupational diseases, venereal diseases/AIDS, and self-rescue in public emergencies. Due to the high risks associated with occupational diseases and AIDS among migrants in China, previous studies on migrant health in China have predominantly concentrated on addressing these specific issues (Gong et al., 2012; Qian et al., 2021). However, there is a lack of research on chronic diseases among migrants, despite their long-term health risks. Our study indicates that health education on chronic diseases has a significant negative impact on migrants’ health status, suggesting the ineffectiveness of current approaches. Chronic diseases pose long-term health risks, and the positive impact of interventions focusing on chronic disease health education has already been established in previous studies (Eckman et al., 2012; Newman, Steed, and Mulligan, 2014). Therefore, efficient education on chronic diseases for the migrants in China could help reduce the risk of chronic diseases that arise from socio-economic disadvantages, thereby contributing to mitigating the chronic disease burden among migrants in China.

                Secondly, traditional educational media, such as lectures and bulletin boards, have a positive impact on migrants’ health status. Considering the effectiveness of face-to-face education as indicated by previous studies (Flora, Maibach, and Maccoby, 1989; Fnnegan Jr and Viswanath, 2008; Yoshida et al., 2012), the positive effects of lectures become evident. However, there is a study showing that low health literacy and poor health status is also related to the limited utiilization of online health education (Chen et al., 2020). Therefore, there is room for online health education to develop, indicating the need for further exploration in online health education.

                Thirdly, the effectiveness of health education varies among different genders and age groups of migrants. In terms of gender, both male and female migrants show significant results, but the impact of health education on the health of female migrants is notably more significant. As mentioned in Section 4, the overall health status of female migrants tends to be lower than that of male migrants. Therefore, when addressing the health needs of the mi-grants in China, the quality of services should be adjusted to account for the quality of life of female migrants. As for age, health education is more likely to yield positive health status for older migrants. This finding indicates that, on the one hand, public health education can improve the health status of older migrants. On the other hand, it may not be as effective in improving the health of younger migrants, particularly those in their 30s. Targeted public health education for younger migrants can help identify and prevent diseases at an early stage. As mentioned in Section 4, despite the need to analyze and differentiate the health effects of public health education on the migrants based on gender and age, considering the socio-economic status of migrants and migration trends, there is a scarcity of relevant research. Therefore, more studies ad-dressing the health effects considering the gender and age of migrants should be conducted.

                Lastly, this study revealed that health behaviors mediate the relationship between health education and the health status of migrants, as supported by the bootstrap test. Therefore, modifying the health behaviors of migrants can lead to improved health status. These findings align with health behavior theories, such as the Health Belief Theory, which propose that health knowledge enhances health behaviors and health status. However, further research is needed to explore the specific mechanisms applicable to the migrant population in China.

                We have provided several policy recommendations. Firstly, to enhance the quality of community-based public health education programs, the Chinese government should encourage their development and innovation. It is essential to develop a comprehensive public health knowledge system based on health behavior theories for universities, students in relevant departments, and practitioners in the field. Secondly, the focus should expand beyond infectious diseases to include non-communicable diseases, especially chronic diseases. Addressing the challenge of educating migrants on chronic diseases is crucial. Thirdly, particular attention should be given to the health needs of female migrants and elderly migrants, as they exhibit higher sensitivity to public health education. Targeted health education for young migrants is essential for early disease prevention. Lastly, traditional media should be prioritized for health education due to its accessibility and minimal technical issues, while efforts should be made to improve the accessibility of online health education programs for migrants in China.

                We have described the strengths and limitations of this study. To avoid redundancy, a detailed analysis of these aspects will be provided in the response 6 to point 6.

Point 6: It would have been better if the researchers had outlined the limitations of the current research.

Response 6: (p. 25) In the discussion section, we have not only outlined the strengths of this study but also addressed its limitations and provided suggestions for future research. This study has two major limitations. Firstly, the impact of health education on the health status of the migrant population was assessed solely based on self-rated health status (SRHS), which serves as a comprehensive health indicator. Future studies can consider examining the influence of health education on various dimensions of health, including physical, psychological, social, and environmental aspects (Bergner, 1989). Secondly, this study primarily focused on the individual-level effects of health education on migrants in China. Future research can gather information on health education for migrants from a health environment perspective, considering the meso- and macro- levels. Exploring how communities and governments can employ diverse empirical research strategies to strike a balance between cost-effectiveness and the health needs of migrants in China would be valuable. Thirdly, while this study conducted heterogeneity analysis of the effects of health education on the health status of migrants based on gender and age, future studies could explore a broader range of socio-economic factors among the migrants in China. Considering additional socio-economic factors would provide valuable insights into how public health education can effectively target the specific needs of the migrants in China.

Reviewer 2 Report

Thank you very much for the opportunity to review the manuscript titled: Relationships between Health Education, Health Behaviors, and Health Outcomes among Migrants in China: A Cross-Sectional Study based on the China Migrant Dynamic Survey

Below are my comments:

In the abstract, please add information about the study population and explain the abbreviation CMDS

Introduction - please explain the terms: health behaviors and health outcome. Please correct the study aim - the sentences are repeated.

Please introduce the study population's flow chart and the study population's inclusion and exclusion criteria. Please describe the study population in a separate section.

Please add information about the reliability of the SRHS in your study.

Please describe in detail the statistical methods also the program you used for the calculations. Table 1. Statistical Description - please check the data in observation. In my opinion, it's impossible that there is the same number.  There is no discussion - no comparison of your work with that of others. Also, please add information about strengths and weaknesses of your study

Author Response

Response to Reviewer 2 Comments

Point 1: In the abstract, please add information about the study population and explain the abbreviation CMDS.

Response 1: We have included additional details in the second and third sentences of the abstract regarding the selected study population and the specific name of CMDS. A total of 169,989 samples were selected as the study population. The precise name of the survey dataset used is China Migrants Dynamic Survey (CMDS).

Point 2: In the introduction section, please explain the terms: health behaviors and health outcome. Please correct the study aim - the sentences are repeated.

Response 2: (p. 2) We have referenced representative literature in the field to add definitions of health behaviors and health status in the introduction section. Additionally, we have clarified the research objectives and made revisions to eliminate redundant sentences.

                Firstly, upon careful review of the manuscript, it was determined to replace the term “health outcomes” with “health status”. This decision is based on the fact that the health variable used in our analysis is particularly related to self-rated health status (SRHS). Therefore, we would like to inform you of our reconsideration of the terminology selection.

                We offered the following elucidations for the two concepts introduced in the introduction section. Health behaviors refer to “personal attributes such as beliefs, expectations, motives, values, perceptions, and other cognitive elements; personality characteristics, including affective and emotional states and traits; and overt behavior patterns, actions, and habits that relate to health maintenance, to health restoration and to health improvement” (Gochman, 1989, p. 169). Health status is commonly defined as individuals’ self-perceived health (Centers for Disease Control and Prevention, 2022). This multidimensional concept encompasses various aspects such as physical, cognitive, emotional, and social well-being, as well as the presence or absence of disabilities (Stewart, Ware, and Ware Jr, 1992). Health surveys often capture these dimensions to assess an individual’s overall health status. It is often referred to as self-rated health status and has consistently shown its significance in predicting important health outcomes including mortality and morbidity (Dowd and Zajacova, 2007).

                Secondly, we have clarified the research objectives. This study aims to explore the causal relationship between public health education and the health status of migrants in China, specifically investigating the impact of health education on migrants’ health status and whether health behaviors mediate this relationship. By comprehensively analyzing these research questions, this study aims to offer valuable suggestions to improve the effectiveness of public health education methods.

Point 3: Please introduce the study population’s flow chart and the study population’s inclusion and exclusion criteria. Please describe the study population in a separate section.

Response 3: (p. 5) We added section 3.2. Study Population to provide an overview of the study population. In the eighth sentence, we discussed the migration range (inclusion and exclusion criteria) within the migrant population. The migration range of the migrant population encompasses cross-provincial, cross-city, cross-county, and cross-border movements. However, we did not consider cross-county and cross-border movements within the migrant population. We also provided a flow chart of the study population (See Figure 2).

Point 4: Please add information about the reliability of the SRHS in your study.

Response 4: (p. 6) We have added an information about the reliability of self-rated health status (SRHS) in section 3.2.1. Dependent Variable. SRHS is a prominent indicator that allows for the assessment of an individual’s actual health status across various contexts. We explained it as follows: ”The self-rated health status (SRHS) is frequently examined in epidemiological surveys to assess individuals’ overall well-being in terms of social, biological, and psychological health (Wilson and cleary, 1995; Idler and Benyamini, 1997). It serves as an indicator that can be applied across various contexts and can be used as a proxy for actual health status (Williams, Di Nardo, and Berma, 2017). Therefore, SRHS can be utilized as a measure to assess the health status of migrants in China.”

Point 5: Please describe in detail the statistical methods also the program you used for the calculations. Table 1. Statistical Description - please check the data in observation. In my opinion, it’s impossible that there is the same number.

Response 5: (p. 5 and p. 8) We conducted quantitative analysis utilizing the CMDS 2017 database, which contains survey items related to public health education. A total of 169,989 samples were extracted from the CMDS 2017 database. IBM SPSS Statistics version 29 was employed as the analysis program, which functions as a random number generator software. In Section 3.2, Study Population, we indicated the analysis program and the year of the database used.

                Regarding data observation, in Table 1, the fact that the observations for the seven health education variables by topic and the five education medium variables are the same is because each variable is not a single categorical variable, but rather a separate dummy variable. In other words, there are separate dummy variables for the seven health education topics and the five education mediums. Therefore, there is no difference in the observations, only the composition of 0 and 1 varies. In the 3.2.2. Independent Variables section, it explicitly states the following about this: “These seven types of health education included occupational disease, venereal dis-ease/AIDS, reproductive health, mental health, chronic health, maternal and child health, and self-rescue in public emergencies. Each type was transformed into a dummy variable to investigate the relationship between the type of disease prevention education and the health outcomes of migrants. The third major independent variable was the type of education medium. There were five types of education medium, including lectures, publicity material, bulletin boards, public consultations, and online education. Each type was transformed into a dummy variable to explore the association between the type of education medium and the health outcomes of migrants.”

Point 6: There is no discussion - no comparison of your work with that of others. Also, please add information about strengths and weaknesses of your study.

Response 6: (p. 23-25) Based on the primary findings, we have significantly augmented the content of the discussion section. This section now encompasses the establishment of associations with previous studies, the exploration of policy implications, and the strengths and limitations of the current study. The revised content includes the following aspects. For a more detailed discussion, please refer to our manuscript.

                We discussed our research findings in relation to previous studies in the field. Firstly, health education has a significant positive impact on the health status of migrants in China, particularly in relation to occupational diseases, venereal diseases/AIDS, and self-rescue in public emergencies. Due to the high risks associated with occupational diseases and AIDS among migrants in China, previous studies on migrant health in China have predominantly concentrated on addressing these specific issues (Gong et al., 2012; Qian et al., 2021). However, there is a lack of research on chronic diseases among migrants, despite their long-term health risks. Our study indicates that health education on chronic diseases has a significant negative impact on migrants’ health status, suggesting the ineffectiveness of current approaches. Chronic diseases pose long-term health risks, and the positive impact of interventions focusing on chronic disease health education has already been established in previous studies (Eckman et al., 2012; Newman, Steed, and Mulligan, 2014). Therefore, efficient education on chronic diseases for the migrants in China could help reduce the risk of chronic diseases that arise from socio-economic disadvantages, thereby contributing to mitigating the chronic disease burden among migrants in China.

                Secondly, traditional educational media, such as lectures and bulletin boards, have a positive impact on migrants’ health status. Considering the effectiveness of face-to-face education as indicated by previous studies (Flora, Maibach, and Maccoby, 1989; Fnnegan Jr and Viswanath, 2008; Yoshida et al., 2012), the positive effects of lectures become evident. However, there is a study showing that low health literacy and poor health status is also related to the limited utiilization of online health education (Chen et al., 2020). Therefore, there is room for online health education to develop, indicating the need for further exploration in online health education.

                Thirdly, the effectiveness of health education varies among different genders and age groups of migrants. In terms of gender, both male and female migrants show significant results, but the impact of health education on the health of female migrants is notably more significant. As mentioned in Section 4, the overall health status of female migrants tends to be lower than that of male migrants. Therefore, when addressing the health needs of the mi-grants in China, the quality of services should be adjusted to account for the quality of life of female migrants. As for age, health education is more likely to yield positive health status for older migrants. This finding indicates that, on the one hand, public health education can improve the health status of older migrants. On the other hand, it may not be as effective in improving the health of younger migrants, particularly those in their 30s. Targeted public health education for younger migrants can help identify and prevent diseases at an early stage. As mentioned in Section 4, despite the need to analyze and differentiate the health effects of public health education on the migrants based on gender and age, considering the socio-economic status of migrants and migration trends, there is a scarcity of relevant research. Therefore, more studies ad-dressing the health effects considering the gender and age of migrants should be conducted.

                Lastly, this study revealed that health behaviors mediate the relationship between health education and the health status of migrants, as supported by the bootstrap test. Therefore, modifying the health behaviors of migrants can lead to improved health status. These findings align with health behavior theories, such as the Health Belief Theory, which propose that health knowledge enhances health behaviors and health status. However, further research is needed to explore the specific mechanisms applicable to the migrant population in China.

                We have provided several policy recommendations. Firstly, to enhance the quality of community-based public health education programs, the Chinese government should encourage their development and innovation. It is essential to develop a comprehensive public health knowledge system based on health behavior theories for universities, students in relevant departments, and practitioners in the field. Secondly, the focus should expand beyond infectious diseases to include non-communicable diseases, especially chronic diseases. Addressing the challenge of educating migrants on chronic diseases is crucial. Thirdly, particular attention should be given to the health needs of female migrants and elderly migrants, as they exhibit higher sensitivity to public health education. Targeted health education for young migrants is essential for early disease prevention. Lastly, traditional media should be prioritized for health education due to its accessibility and minimal technical issues, while efforts should be made to improve the accessibility of online health education programs for migrants in China.

               We have described the strengths and limitations of this study. Firstly, the impact of health education on the health status of the migrant population was assessed solely based on self-rated health status, which serves as a comprehensive health indicator. Future studies could consider examining the influence of health education on various dimensions of health, including physical, psychological, social, and environmental aspects (Bergner, 1989). Secondly, this study primarily focused on the individual-level effects of public health education on migrants in China. Future research could gather information on health education for migrants from a health environment perspective, considering the meso and macro levels. Exploring how communities and governments can employ diverse empirical research strategies to strike a balance between cost-effectiveness and the health needs of migrants in China would be valuable. Thirdly, while this study conducted heterogeneity analysis of the effects of public health education on the health status of migrants based on gender and age, future research could explore a broader range of socio-economic factors among the migrants in China. Considering additional socio-economic factors would provide valuable insights into how public health education can effectively target the specific needs of the migrants in China.

Reviewer 3 Report

This study reports the correlation between health education, health behavior, and health outcomes among immigrants in China. Although the topic of this study is important in terms of public health, the quality of this manuscript needs further improvement.

First of all, the journal guideline must be followed. According to the instructions of this journal, an abstract should not exceed 200 words. However, the abstract of this manuscript appears to be considerably longer than that.

This manuscript does not follow the structure specified in the journal guideline.

It is necessary to clarify the meaning of 2. Literature Review. In other words, if this content corresponds to the rationale of this study, it should be regarded as an Introduction section, and if it corresponds to the method of this study, it should be regarded as a Method section. And, the methods of literature review are unclearly described.

A detailed explanation of health education, which is the variable of interest in this study, is needed. That is, did the participants in this study receive the homogenious health education? The authors classified the education by its topics.

Do the education belonging to the same category mean education of the same content and duration? If the homogeneity of the education is not guaranteed, can it be regarded as a single variable called 'health education'?

The methods and results of this study should be further summarized, and the consideration further increased.

Please add ethical considerations in the conduct of this study.

Please add the unit of houshold income from the table.

The discussino sections are very short compared to their findings. For example, even potential causes explaining differences in effects by gender or age among the main findings they found were not considered in this manuscript.

Others
From the title and abstract of this manuscript, it is difficult to infer the duration of this cross-sectional study series. Therefore, it is desirable to indicate the study period of this study in the title and/or abstract.
The abbreviation CMDS is not defined in the Abstract.
In the Introduction, the abbreviation COVID-19 is not defined.

Author Response

Response to Reviewer 3 Comments

Point 1: The journal guideline must be followed. According to the instructions of this journal, an abstract should not exceed 200 words. However, the abstract of this manuscript appears to be considerably longer than that.

Response 1: According to the journal guidelines, we have adjusted the word count of the abstract in this manuscript to a maximum of about 200 words. The current word count of the abstract is 208 words.

Point 2: This manuscript does not follow the structure specified in the journal guideline. It is necessary to clarify the meaning of 2. Literature Review. In other words, if this content corresponds to the rationale of this study, it should be regarded as an Introduction section, and if it corresponds to the method of this study, it should be regarded as a Method section. And, the methods of literature review are unclearly described.

Response 2: (p. 2-3) The comments regarding the structure of our manuscript likely pertain to the Literature Review section, which is located in section 2 of our manuscript.

                To ensure that the purpose and significance of the literature review are clearly conveyed, we have added additional explanations in our manuscript. To justify the investigation of the impact of public health education on the health status of migrants in China and understand the underlying mechanisms, this study conducted a detailed review of relevant previous studies. Instead of adopting a systematic review approach, a comprehensive review of representative literature was undertaken. This study addressed limitations identified from a limited number of previous studies and established a theoretical foundation to analyze the relationship mechanism between health education and the health status of migrants in China.

Point 3: A detailed explanation of health education, which is the variable of interest in this study, is needed. That is, did the participants in this study receive the homogenious health education? The authors classified the education by its topics. Do the education belonging to the same category mean education of the same content and duration? If the homogeneity of the education is not guaranteed, can it be regarded as a single variable called ‘health education’?

Response 3: (p. 6) In the CMDS dataset, the public health education items refer to the survey on public health education programs, which is one of the National Basic Public Health Services offered to the entire population through community-based initiatives implemented in local villages since 2009 (Jia, 2022). The program themes and content remain consistent across regions. Additionally, for the CMDS data in 2017, the survey on public health education was limited to the past year. The independent variables in this study consist of general public health education variables (treated as continuous variables) and public health education variables categorized by subject (represented by 7 dummy variables). Since the content of public health education is consistent and the time frame is controlled, it is feasible to represent the public health education variable as a single variable or multiple topic-specific variables.

Point 4: The methods and results of this study should be further summarized, and the consideration further increased.

Response 4: (p. 23-25) We further summarized the results of this study and enhanced the discussion in the discussion section. This section now encompasses the establishment of associations with previous studies, the exploration of policy implications, and a comprehensive analysis of the strengths and limitations of the current study. The revised content includes the following aspects. For a more detailed discussion, please refer to our manuscript.

                We discussed our research findings in relation to previous studies in the field. Firstly, health education has a significant positive impact on the health status of migrants in China, particularly in relation to occupational diseases, venereal diseases/AIDS, and self-rescue in public emergencies. Due to the high risks associated with occupational diseases and AIDS among migrants in China, previous studies on migrant health in China have predominantly concentrated on addressing these specific issues (Gong et al., 2012; Qian et al., 2021). However, there is a lack of research on chronic diseases among migrants, despite their long-term health risks. Our study indicates that health education on chronic diseases has a significant negative impact on migrants’ health status, suggesting the ineffectiveness of current approaches. Chronic diseases pose long-term health risks, and the positive impact of interventions focusing on chronic disease health education has already been established in previous studies (Eckman et al., 2012; Newman, Steed, and Mulligan, 2014). Therefore, efficient education on chronic diseases for the migrants in China could help reduce the risk of chronic diseases that arise from socio-economic disadvantages, thereby contributing to mitigating the chronic disease burden among migrants in China.

                Secondly, traditional educational media, such as lectures and bulletin boards, have a positive impact on migrants’ health status. Considering the effectiveness of face-to-face education as indicated by previous studies (Flora, Maibach, and Maccoby, 1989; Fnnegan Jr and Viswanath, 2008; Yoshida et al., 2012), the positive effects of lectures become evident. However, there is a study showing that low health literacy and poor health status is also related to the limited utiilization of online health education (Chen et al., 2020). Therefore, there is room for online health education to develop, indicating the need for further exploration in online health education.

                Thirdly, the effectiveness of health education varies among different genders and age groups of migrants. In terms of gender, both male and female migrants show significant results, but the impact of health education on the health of female migrants is notably more significant. As mentioned in Section 4, the overall health status of female migrants tends to be lower than that of male migrants. Therefore, when addressing the health needs of the mi-grants in China, the quality of services should be adjusted to account for the quality of life of female migrants. As for age, health education is more likely to yield positive health status for older migrants. This finding indicates that, on the one hand, public health education can improve the health status of older migrants. On the other hand, it may not be as effective in improving the health of younger migrants, particularly those in their 30s. Targeted public health education for younger migrants can help identify and prevent diseases at an early stage. As mentioned in Section 4, despite the need to analyze and differentiate the health effects of public health education on the migrants based on gender and age, considering the socio-economic status of migrants and migration trends, there is a scarcity of relevant research. Therefore, more studies ad-dressing the health effects considering the gender and age of migrants should be conducted.

                Lastly, this study revealed that health behaviors mediate the relationship between health education and the health status of migrants, as supported by the bootstrap test. Therefore, modifying the health behaviors of migrants can lead to improved health status. These findings align with health behavior theories, such as the Health Belief Theory, which propose that health knowledge enhances health behaviors and health status. However, further research is needed to explore the specific mechanisms applicable to the migrant population in China.

                We have provided several policy recommendations. Firstly, to enhance the quality of community-based public health education programs, the Chinese government should encourage their development and innovation. It is essential to develop a comprehensive public health knowledge system based on health behavior theories for universities, students in relevant departments, and practitioners in the field. Secondly, the focus should expand beyond infectious diseases to include non-communicable diseases, especially chronic diseases. Addressing the challenge of educating migrants on chronic diseases is crucial. Thirdly, particular attention should be given to the health needs of female migrants and elderly migrants, as they exhibit higher sensitivity to public health education. Targeted health education for young migrants is essential for early disease prevention. Lastly, traditional media should be prioritized for health education due to its accessibility and minimal technical issues, while efforts should be made to improve the accessibility of online health education programs for migrants in China.

               We have described the strengths and limitations of this study. Firstly, the impact of health education on the health status of the migrant population was assessed solely based on self-rated health status, which serves as a comprehensive health indicator. Future studies could consider examining the influence of health education on various dimensions of health, including physical, psychological, social, and environmental aspects (Bergner, 1989). Secondly, this study primarily focused on the individual-level effects of public health education on migrants in China. Future research could gather information on health education for migrants from a health environment perspective, considering the meso and macro levels. Exploring how communities and governments can employ diverse empirical research strategies to strike a balance between cost-effectiveness and the health needs of migrants in China would be valuable. Thirdly, while this study conducted heterogeneity analysis of the effects of public health education on the health status of migrants based on gender and age, future research could explore a broader range of socio-economic factors among the migrants in China. Considering additional socio-economic factors would provide valuable insights into how public health education can effectively target the specific needs of the migrants in China.

Point 5: Please add ethical considerations in the conduct of this study.

Response 5: (p. 25) The use of China Migrants Dynamic Survey 2017 data did not require any additional ethical approval, as ethical approval was already granted by the Ethics Review Board of the National Health Commission for the China Migrants Dynamic Survey 2017 data. Written informed consent was obtained from all participants as well. Since our study involved the analysis of de-identified existing data, no further ethical approval was necessary. We have included an Institutional Review Board Statement section in the final part of our manuscript, where we have provided a statement regarding the ethical approval of the research.

Point 6: Please add the unit of household income from the table.

Response 6: We have included the unit (CNY) for household income in all the presented analysis results tables.

Point 7: From the title and abstract of this manuscript, it is difficult to infer the duration of this cross-sectional study series. Therefore, it is desirable to indicate the study period of this study in the title and/or abstract.

Response 7: We have indicated the year of the data used for analysis in the abstract of this paper. This study examined the effects of public health education on the health status of migrants in China, utilizing cross-sectional data from the China Migrants Dynamic Survey (CMDS) conducted in 2017.

Point 8: The abbreviation CMDS is not defined in the Abstract. In the Introduction, the abbreviation COVID-19 is not defined.

Response 8: (p. 2) In the second sentence of the abstract, we explicitly mentioned the original database name, ‘China Migrants Dynamic Survey’, instead of using the abbreviation CMDS. In the second paragraph of the introduction section, we have updated 'COVID-19' to its original term, ‘Coronavirus disease 2019’.
